# Antitumoral Properties of the Nutritional Supplement Ocoxin Oral Solution: A Comprehensive Review

**DOI:** 10.3390/nu12092661

**Published:** 2020-08-31

**Authors:** Atanasio Pandiella-Alonso, Elena Díaz-Rodríguez, Eduardo Sanz

**Affiliations:** 1Instituto de Biología Molecular y Celular del Cáncer, CSIC-IBSAL and CIBERONC, 37007 Salamanca, Spain; ediaz@usal.es; 2Catalysis S.L., 28016 Madrid, Spain; eduardo@catalysis.es

**Keywords:** ocoxin, antitumoral, antioxidant, natural product, nutritional supplement

## Abstract

Ocoxin Oral Solution (OOS) is a nutritional supplement whose formulation includes several plant extracts and natural products with demonstrated antitumoral properties. This review summarizes the antitumoral action of the different constituents of OOS. The action of this formulation on different preclinical models as well as clinical trials is reviewed, paying special attention to the mechanism of action and quality of life improvement properties of this nutritional supplement. Molecularly, its mode of action includes a double edge role on tumor biology, that involves a slowdown in cell proliferation accompanied by cell death induction. Given the safety and good tolerability of OOS, and its potentiation of the antitumoral effect of other standard of care drugs, OOS may be used in the oncology clinic in combination with conventional therapies.

## 1. Introduction

Ocoxin (also termed Oncoxin) Oral Solution (OOS) is a nutritional supplement that includes several plant and natural products with an ample spectrum of biological activities. Some of these products have antioxidant or antitumoral properties. The rationale behind the use of OOS is double. On the one hand, persons suffering certain pathologies or aging may benefit from the supplementation of products included in OOS. On the other hand, the known favorable properties of the constituting compounds on a variety of cellular functions may promote an increase in the therapeutic effect of drugs that are taken by the patients with certain diseases, and therefore increase their quality of life. In fact, as will be discussed below, a combination of standard of care drugs used in the oncology clinic with OOS has been reported to increase the therapeutic effect of the former. Another important aspect of OOS as a nutritional supplement is its large therapeutic index. In fact, the recommended dosage has not caused noticeable toxicities. In this review, we describe OOS constituents with particular attention to their antitumoral properties. We also present preclinical and clinical evidence of the antitumoral action of OOS, and finally, describe the mechanism of this effect.

## 2. OOS Components

OOS is a formulation that includes several components (Table 1); some of them have substantial literature supporting their antitumoral properties. Below we describe some of the reported evidence of the effect of the different constituents of OOS on tumoral cells.

### 2.1. Plant Extracts

#### 2.1.1. Green Tea Extract

Green tea is produced from the leaves of *Camellia sinensis*. It is widely consumed in the world, especially in Asia and in the Middle East [1]. A number of studies have shown the beneficial properties of green tea on frequent diseases such as obesity, neurodegenerative disorders, inflammatory diseases, and cancer [2]. An important component of green tea leaves is epigallocatechin (EGC). The initial step in the action of EGC likely involves interaction with a cellular protein that acts as a receptor. Thus far, EGC has only been reported to bind to the 67-kDa laminin receptor [2], a non-integrin cell surface receptor. It is unclear whether other proteins may also act as cellular receptors for EGC.

An ample number of reports have explored the mechanisms of the antiproliferative action of ECG [3]. EGC appears to have at least a double antitumoral effect: it impairs cell cycle progression and favors cell death (Figure 1). With respect to the former, cytometric studies in liver cancer cells showed that EGC caused inhibition of G1 to S cell cycle progression [4]; however, EGC can also affect cell cycle progression by acting on other cell cycle phases [2]. Biochemically, ECG reduced expression of several cyclins, including cyclin D1. EGC also induced dephosphorylation of the retinoblastoma protein, which may explain the G1 to S blockade [3]. EGC has also been reported to increase the level of inhibitors of cell cycle progression such as p27, p21, p16, and p18 [2]. In addition to effects on these cell cycle regulatory proteins, EGC can also affect cytokinesis. This latter process is critical during mitosis, allowing the final division of the cell to create two daughter cells. EGC, through binding to its 67-kDa laminin receptor, whose expression is increased in several cancers, has been shown to affect the function of myosin II regulatory light chain by reducing its phosphorylation at residues critical for its action, which are required for proper cytokinesis [5].

EGC has also been reported to affect signaling by growth factor receptors such as the epidermal growth factor receptor, the insulin-like growth factor receptor I or the vascular endothelial growth factor receptor [6]. Action on the latter may explain the antiangiogenic actions of EGC, which contribute to its antitumoral effect [7]. One of the target pathways affected by EGC is the ERK1/2 mitogen activated protein kinase [8,9]. That pathway is a critical player in the transduction of proliferation responses by growth factor receptors [10]. Another important action of EGC relates to its effect on cancer stem cells, restricting their proliferation and therefore affecting stemness in several malignancies, including breast [11], lung [12], or colorectal cancer [13]. Mechanistically, EGC acted through a cGMP-FOXO3 axis. FOXO3 is a transcription factor involved in several biological actions including the control of stemness and also acts as a tumor suppressor [2].

A relevant effect of EGC refers to its action on epigenetic modulators that control DNA methylation and histone acetylation status. DNA methyltransferases (DNMTs) and histone deacetylases (HDACs) participate in transcriptional gene silencing, while histone acetyl transferases (HATs) positively regulate gene expression. EGC can bind to DNMT, inhibiting its enzymatic activity [9] and causing the reactivation of methylation-silenced genes in prostate cancer PC3 cells [14]. In skin cancer cells, EGC decreased DNMT1, DNMT3A, and DNMT3B activity and expression, and increased acetylation of histones H3 and H4. The latter can be attributed to EGC acting as a histone deacetylase inhibitor. These epigenetic effects were claimed as responsible for upregulation of the cell cycle inhibitors p16 and p21 [15].

Several studies have also investigated the apoptotic and autophagic actions of EGC. In HepG2 hepatocarcinoma and PC12 pheochromocytoma cells, EGC augmented the proapoptotic protein BAX and down-regulated the antiapoptotic BCL-2 [16]. The net result was an increase in the BAX/BCL-2 proteins that facilitated apoptotic cell death. EGC may also control autophagic mechanisms through mTOR, either promoting or inhibiting cell death depending on the tumor cell type [3]. Another relevant action of EGC is the regulation of oxidative stress and inflammation through its effect on cycloxigenase-2 (COX-2). EGC has been shown to inhibit COX-2 activity as well as decrease COX-2 levels [3]. Moreover, EGC has been reported to decrease the activity of ERK1/2 and p38, which are upstream of COX-2 [8]. Those authors reported that in skin cancer, EGC exerted antitumoral activity mediated by MAPK suppression, which decreased COX-2 activity. Conversely, treatment of human chondrocytes with EGC inhibited COX-2 and iNOS activity, indicating a possible involvement of this treatment in the inhibition of cartilage resorption in arthritis [17].

#### 2.1.2. Cinnamon Extract

*Cinnamomum cassia* extracts contain several active components such as the essential oils cinnamic aldehyde and cinnamyl aldehyde, tannin, eugenol, β-caryophyllene, and its oxidized form [18]. Among the multiple mechanisms by which cinnamon extract (CE) exerts antitumoral properties, it is worth mentioning the proapoptotic actions of the constituents present in the extract (Figure 1). In promyelocytic leukemia [19] as well as in colon cancer cells [20], eugenol triggered apoptosis through induction of DNA damage with fragmentation as well as increased production of reactive oxygen species (ROS), which was accompanied by loss of mitochondrial membrane potential. DNA fragmentation leads, in turn, to caspase 3 and caspase 9 cleavage, which are indicators of apoptotic cell death. In a model of skin carcinogenesis, eugenol increased the BAX/BCL-2 ratio and increased caspase 3 cleavage [21]. Moreover, eugenol increased the effect of the chemotherapeutic drug gemcitabine on HeLa cells, synergistically amplifying the effect on caspase 3 activity with respect to each agent alone [22]. This finding is relevant as it indicates that CE compounds may increase the action of at least some cytotoxic compounds.

Cinnamaldehyde, another component found in CE, also induced DNA damage and apoptosis in HL60 promyelocytic leukemia cells. Furthermore, it induced clear hallmarks of apoptotic cell death, including cleavage of caspases 3 and 9, and loss of mitochondrial membrane potential that was accompanied by cytochrome C release to the cytosol [23]. Even at low doses (1 μM) cinnamaldehyde has been shown to promote apoptosis through increases in caspase 3, BAX, and BID activities, accompanied by the down-regulation of prosurvival factors MCL-1 and BCL-2 in the hepatoma cell line PLC/PRF/5 [24]. Other cinnamaldehyde derivatives found in CE, such as hydroxycinnamaldehyde and 2-benzoyloxycinnamaldehyde, also show antitumoral activity by promoting cell death. The former has been reported to induce apoptosis by decreasing NFκB, and by down-regulating the transcription factors c-fos and c-jun [25].

Besides the action of CE components on apoptotic mechanisms, CE exerts antitumoral properties through regulation of cell cycle progression. In breast and colon cancer cells, CE components increase cyclin B and provoke stalling of the cell cycle at G2/M [26]. In cellular models of these diseases, CE inhibits signaling by the AKT and STAT3 routes, which are involved in cell growth and survival pathways.

#### 2.1.3. Glycyrrhiza Glabra Root Extract

One of the most relevant components of *Glycyrrhiza glabra* extract, from a medicinal plant point of view, is glycyrrhizin. There is an ample list of papers describing the antitumoral effects of this compound, including effects on cell proliferation and induction of programmed cell death [27] (Figure 1). One of the actions of glycyrrhizin is reduction of stemness in hepatocellular carcinoma by acting on the Jun kinase pathway [28]. In fact, glycyrrhicic acid directly interacts with Jun kinase 1 and, through the activation of this kinase, inhibits stemness of hepatocarcinoma cells. In this latter pathology, glycyrrhicic acid appears to have therapeutic and chemopreventive roles. Thus, glycyrrhicic acid has been reported to inhibit degeneration of viral hepatitis B into hepatocarcinoma [29], supporting a role of this compound as a chemopreventive treatment in chronic hepatitis B [30].

Glycyrrhicic acid has also been involved in the control of metastatic dissemination. Thus, in prostate cancer cells, glycyrrhicic acid affected epithelial-mesenchymal transition through regulation of high mobility group box 1 (HMGB1) proteins [31]. These proteins facilitate cell migration and invasion. In fact, in the prostate cancer cell line DU145, glycyrrhicic acid inhibited, in a dose-dependent manner, cell migration capacity due to the alteration in epithelial to mesenchymal transition. In another report, Qiu et al. demonstrated that glycyrrhizic acid prevented high-fat diet-promoted formation of a metastatic niche [32]. Glycyrrhizic acid decreased metastatic spread in several preclinical models including the 4T1 breast cancer and B16F10 melanoma cell lines and that was accompanied by gut microbiota disbiosis. Therefore, glycyrrhizic acid-mediated control of gut microbiota may also promote an anti-metastatic phenotype.

One of the interesting aspects that remains to be explored is the careful analysis of the antitumoral effect of the combination of the three major components of OOS, i.e., the combination of Glycyrrhiza glabra extract, green tea extract, and cinnamon extract.

### 2.2. Vitamins

#### 2.2.1. Ascorbic Acid

In addition to its role as a cofactor for several biosynthetic and gene regulatory enzymes, ascorbic acid (vitamin C) is well-known for its antioxidant properties, which may help in controlling oxidative stresses linked to cancer generation/progression (Figure 1). Moreover, vitamin C is an important contributor to immune defense of the body and has been found to be deficient in patients with advanced stages of cancer [33]. With independence of its mechanism of action, a large number of reports have shown that pharmacological concentrations of vitamin C can selectively kill cancer cells in vitro [34]. Moreover, some reports have also demonstrated that vitamin C can inhibit growth of a number of human tumor xenografts in immunodeficient mice. In addition, some reports have also supported a chemopreventive action of vitamin C [33,34]. One of the relevant actions of vitamin C is its effect on cancer stem cells. Using a metabolomic approach, Agathocleous et al. observed that ascorbate depletion cooperated with mutations in the receptor tyrosine kinase Flt-3 to promote a leukemogenic phenotype, reversed by dietary ascorbate [35].

#### 2.2.2. B Vitamins

OOS includes the following B complex vitamins: pantothenate (vitamin B5), pyridoxine (vitamin B6), folic acid (vitamin B9), and cyanocobalamin (vitamin B12). These vitamins are important enzyme cofactors or cofactor precursors. B group vitamins including pantothenic acid can participate in modulation of various epigenomes (Figure 1), suggesting that their combination with epigenetic drugs may potentiate their action. There are numerous papers that provide evidences of the antitumoral properties of pyridoxine [36,37]. Moreover, vitamin B6 has been recently reported to increase the antitumoral effectiveness of chemotherapy in breast cancer [38].

Folic acid acts as a co-enzyme in the form of tetrahydrofolate which is involved in pyrimidine nucleotide synthesis. As the latter is required for normal and tumoral cell division, chemotherapeutic agents targeting this metabolic route have been developed for its use in the oncology clinic. This is particularly relevant in colon cancer in which chemotherapy using antifolates has represented a standard of care [39].

The critical role of vitamin B12 in the formation of blood cells makes it an attractive nutritional supplement for patients receiving myelosuppressive therapies or myeloablative procedures. Bone marrow transplantation is frequently used for the therapy of several haemato-oncological malignancies and the recovery of normal hematopoiesis is required after these treatments and could be enhanced by this vitamin. On the other hand, the particular sensitivity of the blood forming system to antitumoral agents also appears an opportunity for the use of supplements that may help in boosting normal hematopoiesis [40].

#### 2.2.3. Amino Acids

Amino acids represent the building blocks of proteins [41]. In addition to this basic structural function, some of them contribute to important physiological processes, as in nutrient sensing through the mTOR route [42], or acting as neurotransmitter as is the case of glutamate [43]. OOS includes several amino acids that in addition to acting as nutritional supplements have also been described to play a role in the control of cell growth.

#### 2.2.4. Glycine

Glycine comprises up to 10% of the total body amino acids. Its beneficial effect for human health has been reported in several pathologies. In fact, glycine has been shown to benefit patients with arthritis, gastric ulcer, hemorrhagic shock, hepatotoxicity, or after organ transplantation [44]. In addition, glycine presents beneficial effects in cancer. Thus, dietary glycine has been shown to inhibit cell proliferation caused by WY-14,643, which is a peroxisome proliferator-activated receptor alpha (PPARα) agonist [45]. Moreover, synthesis of tumor necrosis factor-α (TNF𝛼) by liver Kupffer cells and activation of nuclear factor 𝜅B (NFκB), which may promote tumor expansion, are blocked by glycine. Glycine has also been reported to inhibit tumor growth of implanted B16 melanoma cells [46]. On the other hand, some reports have indicated that dietary restriction of serine and glycine may extend lifespan of animal models with lymphoma or intestinal tumors [47].

#### 2.2.5. Arginine

Arginine plays an important role in regular cellular functions. In fact, the presence of arginine is needed by tumoral cells for growth, and that circumstance has promoted the concept of using diets in which arginine deprivation could be beneficial for patients [48]. However, arginine has also been reported to represent a critical component of body systems for keeping in check abnormal cells such as tumor ones [49]. Thus, arginine is an important component of immune surveillance (Figure 1). This system is critically involved in the targeting and elimination of malignant cells by the organism. In animal models of cancer, arginine supplementation delayed tumor growth and prolonged survival [50].

Arginine-derived nitric oxide has many overlapping and even contradictory roles in tumor initiation, promotion, and progression. Thus, arginine-derived nitric oxide production may induce apoptosis of tumoral cells or increase the effectiveness of radiation therapies [51]. Of particular interest is the role of arginine-derived nitric oxide in tumor immunity. In fact, nitric oxide may play a role in tumor cell cytotoxicity by increasing the antitumoral action of several cells of the immune system such as macrophages, natural killer cells, or cytotoxic T-cells [52,53]. As arginine has been reported to be necessary for both cancer growth and normal immune function, several clinical trials are analyzing the role of arginine deprivation versus supplementation in patients with cancer [49].

#### 2.2.6. Cysteine

Cysteine is a sulfur-containing amino acid. While some cysteine derivatives have shown antitumoral activity in cancer, it is likely that its antitumoral action may be related to its role as an antioxidant (Figure 1). In fact, the cysteine prodrug N-acetyl cysteine (NAC) has been used as antioxidant. NAC has been demonstrated to decrease endogenous oxidant levels and to protect cells against several pro-oxidative insults. Mechanistically, it has been shown that NAC-derived cysteine is desulfurated to generate hydrogen sulfide that is then oxidized in mitochondria to sulfane sulfur species [54]. These sulfane sulfur species may be the responsible molecules in mediating the antioxidative and cytoprotective effects provided by NAC.

### 2.3. Sugars

#### Glucosamine and Sucralose

Synthesized from glucose, glucosamine is an endogenous saccharide. It is an intermediate metabolite in the biosynthesis of proteoglycans and glycosaminoglycans, which are essential sugar-protein molecules found in cartilage. Glucosamine is usually marketed as a dietary supplement in combination with chondroitin sulfate for the improvement of joint health, especially arthritis and degenerative joint disease. Of note, glucosamine has anti-inflammatory activity [55], which, besides its beneficial effect on joint health, may also offer protection to cancer, given the role of chronic inflammation on certain types of tumors, such as colon cancer. In fact, glucosamine has been reported to provide protection against colon cancer [56]. A chemopreventive action of glucosamine has also been reported in lung cancer [57]. Mechanistically, the anti-inflammatory activity of glucosamine has been linked to reduction of proinflammatory cytokines such as IL-6 or IL-1β [58].

Sucralose is a zero-calorie artificial sweetener. Recent studies point to the lack of carcinogenicity of this agent [59]. Moreover, in combination with other sweeteners (commercial formulations), could exert anti-inflammatory roles through the modulation of cytokine expression, and could potentially act as a carcinopreventive agent [60].

### 2.4. Other Components of OOS

Zinc sulfate is another component of OOS. Zinc deficiency has been linked to different pathologies, including cancer and its dietary supplementation could contribute to cancer prevention [61]. In fact, its antitumoral effect seems to be mediated by anti-inflammatory, antioxidant, and immunostimulatory actions [62] (Figure 1). Furthermore, in head and neck cancers, a zinc deficit mediates cellular immunity impairment, contributing to a decrease in T lymphocyte numbers and such effect can be reverted by zinc sulfate supplementation [63].

Malic acid is an intermediate of metabolism created during conversion of sugars to energy. Many fruits are also a source of malic acid, particularly apples. While this compound has been used in fibromyalgia rheumatica patients [64], its potential use as a direct regulator of cell proliferation in cancer has not attracted much attention.

Other components of OOS include manganese sulfate, sodium benzoate, or potassium sorbate.

## 3. OOS in Preclinical Models

The use of preclinical models has helped to confirm the antitumoral properties of OOS, and those studies have resulted particularly helpful in defining aspects such as synergistic actions of OOS with standard of care treatments used in the oncology clinic. Moreover, as will be defined in later sections of this review, the in vitro and in vivo preclinical studies allowed uncovering of the mechanisms by which OOS exerts its antitumoral actions.

### 3.1. Breast Cancer

OOS has been mainly analyzed in the HER2+ subtype of breast cancer [65]. This type of tumors represents around 20% of all the breast tumors and is characterized by the presence of elevated levels of the transmembrane tyrosine kinase HER2. Because of the oncogenic properties of this protein, strategies that target it have reached the oncology clinic. Such strategies include antibodies to the extracellular domain of HER2 as well as small molecule inhibitors of the kinase activity [10]. In cellular models of human HER2+ breast cancer, OOS provoked a dose-dependent inhibition of their proliferation [65]. Biochemically, OOS did not affect the tyrosine phosphorylation of HER2, which is required for the oncogenic action of that tyrosine kinase. The most evident action of OOS in these cells was to increase the amount of the negative cell cycle regulator p27. Such an effect is particularly attractive since the mechanism of action of gold standards in the therapy of breast cancer such as the humanized monoclonal antibody trastuzumab or the small molecule inhibitor lapatinib are expected to cause increases in p27 as part of their antitumoral mechanism of action [10]. Interestingly, OOS augmented the antitumoral action of lapatinib, and inhibited tumor growth in in vivo models of human HER2+ breast cancer. In addition to its effect on the cell cycle, OOS also triggered apoptotic cell death, upregulating caspase 3 and caspase 8 cleavage, used as readouts of the activities of these caspases.

The effect of OOS on the cell cycle, as identified in the biochemical [65] and the in silico analyses [66], opens an interesting possibility for its application in a particular subtype of breast cancer. In fact, hormonal breast tumors, characterized by the presence of estrogen and progesterone receptors, are particularly sensitive to the action of CDK4/6 inhibitors. Therefore, it is possible that hormonal breast tumors may also be sensitive to OOS. Such possibility, and a potential synergistic effect of OOS with CDK inhibitors deserves to be explored.

### 3.2. Leukemias

Acute myelogenous leukemia is a hemato-oncological disease whose incidence is rising in developed countries. In cellular models of this type of leukemia, OOS exerted an antitumoral action characterized biochemically by an increase in p27 [67]. The antiproliferative action of OOS was more effective in combination with standard of care drugs used for the treatment of patients with this disease, such as doxorubicin, Ara C, or fludarabine. In vivo studies confirmed the antitumoral action of OOS in mice injected with HEL cells, decreasing the amount of the proliferation marker Ki67 in tumors from mice treated with OOS. Interestingly, OOS increased a number of cytokines in the serum of these mice, including IL-6. These studies were complemented by in vivo microarray analyses of the effect of OOS on the transcriptomic landscape of leukemic cells [66]. They showed that OOS caused deregulation of 150 genes. Of them, 33 genes were deregulated more than two-fold. Pathway analyses of the deregulated genes pointed to the cell cycle as a major deregulated function, a finding that fits with the cell biological and biochemical data. In addition, genes belonging to the EGFR, IGF-IR, and TGFβ receptor signaling pathways were also deregulated by OOS.

### 3.3. Digestive Tract Neoplasias

One of the most frequent oncological pathologies is colorectal carcinoma. Metastatic spreading to the liver represents one of the major life-threatening morbidities of that disease. Using in vitro models of colorectal cancer, it was reported that OOS reduced cell viability of colorectal carcinoma cells [68]. In the in vivo model used to study the metastatic dissemination of tumor cells to the liver, OOS decreased the size of such metastatic lesions and increased apoptotic cell death within the tumors. OOS also reduced the production of proinflammatory mediators such as COX-2, IFNγ, IL1β, and TNFα. Furthermore, when OOS was administered in combination with irinotecan, a drug commonly used in colorectal cancer, a synergistic reduction in metastatic spreading of the tumoral cells to the liver was detected [69].

In hepatocellular carcinoma, OOS reduced proliferation in the HepG2 and Huh7 cell lines in a time- and dose-dependent manner [70]. The antitumoral effect of OOS in this model was mainly due to a reduction of cell proliferation without induction of cell death. In HepG2 cells, the effect on the cell cycle was characterized by an accumulation of pHistone H3, a marker of mitosis, and also increased cyclin B. The effect on these G2/M markers suggest that in these cells OOS caused cell cycle arrest in such phases of the cell cycle. OOS exerted a synergistic effect when combined with the standard of care sorafenib. In vivo studies in mice xenografted with hepatocarcinoma cells showed that OOS decreased tumor growth and caused necrosis. Using luciferized HepG2 cells injected into the liver, it was observed that OOS decreased tumor aggressiveness since control mice required sacrifice sooner than those treated with OOS.

In pancreatic cancer cells, derived from a murine pancreatic carcinoma, OOS increased the antitumoral effect of paclitaxel and gemcitabine, two drugs commonly used in the therapy of this disease [71]. In addition, OOS exerted an inhibitory effect on the chemoresistant properties provided by stromal cells. Studies carried out in mice that were orthotopically injected with 266/6 murine pancreatic adenocarcinoma cells confirmed the antitumoral effects of OOS in that in vivo model.

### 3.4. Lung Cancer

The activity of OOS has also been analyzed in lung cancer, particularly in small cell lung cancer. In GLC-8 and DMS-92 cell lines, OOS caused a dose-dependent decrease in their proliferation, an effect augmented by the standard of care drug vincristine [72]. In mice injected with GLC-8 cells, OOS decreased tumor growth as compared to animals treated with vehicle. In vitro analyses of the antitumoral effect of OOS demonstrated that it caused cell cycle retardation as well as cell death, effects that were also verified in vivo. The latter was characterized by increased levels of active caspases, particularly the effector caspase 3 as well as caspase 9 and caspase 7. As seen in other tumoral cell types, OOS provoked an increase in p27. Microarray analyses of control versus OOS-treated tumors, followed by pathway enrichment studies (GSEA analyses) defined several biological functions altered in the OOS-treated animals. Of particular interest was the deregulation of DNA damage responses, cell cycle regulation and the PI3K/AKT/mTOR routes. GSEA analyses also indicated deregulation of apoptotic routes, cell cycle progression and angiogenesis [72].

### 3.5. Glioblastoma

OOS has been reported to reduce the self-renewal of glioblastoma initiating cells, causing a slow tumor growth [73]. OOS decreased viability as well as the nanosphere forming activity of glioblastoma stem cells. In addition, OOS regulated microenvironmental macrophages. It changed the polarization markers, decreasing those related to immunosuppressive actions, while increasing those associated with an inflammatory phenotype. These data, together, indicate that OOS participates in the regulation of two critical aspects related to the oncogenic phenotype. On the one hand, it reduces cancer stem cells. On the other, it facilitates fitness of the immune system. Moreover, given the increasing relevance of immune-targeted therapies in several types of cancer, the action of OOS on the immune system may favor the efficacy of those immunomodulatory therapies.

## 4. Clinical Activity of OOS

The clinical effectiveness of OOS has been explored in several clinical trials (Table 2). Some of these studies have focused in the impact of OOS in the quality of life of patients. Thus, Chon-Rivas et al. [74] demonstrated that administration of OOS during radiotherapy or chemotherapy in patients with head and neck cancer decreased the toxicity of these treatments, improving quality of life. Moreover, OOS did not interfere with the action of those therapies. Analogous conclusions were reached by Kaidarova et al. [75], who investigated the impact of OOS on 133 patients with gastric or non-small cell lung cancer. The study, registered at clinicaltrials.gov as NCT03550482, used the Edmonton Symptom Assessment System Questionnaire (ESASQ) to evaluate quality of life. In patients receiving OOS the ESASQ was better than the control arm which received chemotherapy only. Moreover, by the end of three weeks, the level of albumin was higher in the OOS cohort and liver toxicity was lower in that cohort. Benefits of OOS in patients treated with chemotherapy or radiotherapy were also observed by Shumsky et al. [76], who studied the impact of OOS on oral mucositis in 15 patients, 10 treated with OOS and five control. Using the World Health Organization Oral Toxicity Scale, the authors reported a 41% decrease in oral mucositis in the OOS-treated group as compared to minimal change in the control group, which was allowed to use other types of care for oral mucositis. At the termination of the study, the difference was even larger, with 73% improvement in the OOS-treated cohort, and a 20% improvement in the control group. Patients in the OOS group were able to eat normally during 65% of the time, while in the control group that characteristic was reduced to 29%. In another study, the potential of OOS in reducing acute toxicity produced by the radio or chemotherapy in patients with cervical cancer was explored [77]. Patients treated with chemotherapy benefited from OOS. In fact, the adverse events of the chemotherapeutic regime were 20.6% less in those patients receiving OOS, as compared to the placebo group. The cohort receiving OOS also experienced better quality of life in terms of social and emotional functions.

In hepatocellular carcinoma patients, Dzhugashvili et al. analyzed the effect of OOS on micronutrient deficiency in cancer patients [78]. That study was based on the fact that some patients may be deficient in certain amino acids, vitamins, or minerals, some of which are included in OOS formulation. In that work, OOS improved both appetite and quality of life. Moreover, the data pointed to an increase in overall survival in patients with advanced hepatocellular carcinoma. In the latter pathology, a study by Al-Mahtab et al. [79] showed a substantial difference on overall survival in patients treated with OOS. In that study, OOS increased length of overall survival in end-stage hepatocellular carcinoma patients.

Gray-Lovío et al. studied the effect of OOS on patients with stage IIB-III of cutaneous melanoma [80]. Compared to current data obtained from the literature, these authors observed a slight increase in overall survival in patients treated with OOS and suggested that the use of OOS could help improving the quality of life of melanoma patients in those disease stages. These authors also performed a study on the efficacy of topical as well as oral administration of OOS for patients bearing actinic keratoses. These pathologies represent pretumoral lesions of the skin, which are considered in situ squamous cell carcinomas. Histological improvement of the actinic keratoses was observed in 75% of the patients receiving OOS. Moreover, other signs of skin photoaging also improved upon treatment with OOS. This study also reported lack of local or systemic adverse effects of OOS.

## 5. Mechanism of Action of OOS

In vitro as well as in vivo preclinical studies demonstrated the antitumoral activity of OOS. Such an effect could be due to a reduction in cell proliferation, an increase in cell death, or a combination of both. The preclinical studies have in fact reported actions of OOS on cell duplication and cell death. The molecular mechanisms by which OOS achieves such action will be described in the next paragraphs, and is schematically shown in Figure 2.

Effect on the cell cycle. Probably one of the most important antiproliferative mechanisms of action of OOS is related to its cell cycle inhibitory properties, likely caused by some essential constituents of OOS (Table 1 and Figure 1). OOS has been reported to increase the amount of the negative cell cycle regulator p27 in a number of different tumoral cells [65,67,72] (Figure 2). In addition, in breast cancer cells of the HER2+ subtype OOS caused the increase in the amount of p27 that was accompanied by a decrease in cyclin D1 and Rb phosphorylation. The latter effect on Rb was also observed in small cell lung cancer cells treated with OOS [72]. In that tumor model, a search for genetic programs regulated by OOS using microarray techniques and GSEA analyses demonstrated down-regulation of cell proliferation genes in tumors treated with OOS as compared to untreated controls. Together, these events are expected to inhibit progression along the cell cycle, especially from G1 to M. In fact, cytometric studies confirmed that OOS inhibited G1 to M progression. In the breast cancer cell models, down-regulation of cyclin D1 was enhanced if OOS was combined with lapatinib [65], supporting that OOS may increase the efficacy of the latter standard of care drug. In hepatocellular carcinoma, addition of OOS provoked an increase in the amount of cyclin B1 that accompanied cell cycle arrest in G2 [70]. An unbiased microarray study performed on in vivo lung cancer and acute myeloid leukemia cells derived from xenografts prepared in mice, followed by functional transcriptomic evaluation revealed the cell cycle-related functions as the major target functions affected by OOS [66]. These studies confirmed the regulation of p27 and the Rb pathway by OOS in both types of xenografted tumors.

In addition to these effects of OOS on proteins that affect progression of the cell cycle through G1, in HepG2 hepatocellular carcinoma cells, OOS provoked accumulation of pHistone H3, a marker of mitosis, and increased cyclin B. The effect on these G2/M markers suggests that OOS may also act on these cell cycle phases in addition to G1. It is also relevant to comment that the effects of OOS on the cell cycle fit with the expected mechanisms of action of several of the constituents of OOS shown in Table 1 and illustrated in Figure 1. In fact, EGC or glycyrrhicic acid have been shown to exert part of their antitumoral action by braking cell cycle progression. It is therefore not surprising that OOS causes cellular action, likely by a mechanism that involves slowing cell cycle progression by acting at different cell cycle phases (Figure 2).

Proapoptotic actions. In HER2+ cells, in addition to its effect on the cell cycle, OOS stimulated cleavage of PARP, a readout of apoptotic cell death, especially at high doses [65]. Furthermore, OOS triggered a DNA damage response, as indicated by phosphorylation of histone H2AX. This latter biochemical event, pH2AX up-regulation, is frequently seen when double strand breaks in the DNA occur. One of these circumstances happens in the case of the DNA damage caused by apoptotic stimuli that cause DNA double strand breaks, a phenomenon aided by the up-regulation of endonuclease activities during apoptosis. In breast cancer cells, pH2AX up-regulation was very much potentiated if OOS was combined with lapatinib. In small cell lung cancer cells, treatment with OOS also resulted in processing of PARP. These studies also demonstrated that OOS provoked cleavage of caspase 3 both in vitro and in vivo, and giving support to the fact that OOS can trigger caspase-dependent cell death [72]. A similar increase in caspase 3 and the subsequent apoptotic cell death has also been described in metastatic liver lesions coming from colorectal carcinoma treated with OOS [68].

Immune system and circulating cytokines. In a physiological in vivo context using mice and a multiplexed assay to measure a number of circulating cytokines, OOS significantly increased the amount of IL-6 [72]. In contrast, in a colorectal carcinoma model of metastasis to the liver, OOS treatment caused decrease in that cytokine in the liver tissue [68]. In that same study, OOS also reduced IFNα, TNFα, IL-1β, and COX2. In an ulterior study, the same authors added IL-12 and IFNγ to the list of cytokines down-regulated by OOS, while IL-10 was up-regulated [71].

Regulation of stemness. The action of OOS in the regulation of tumoral stem cells has been analyzed in glioblastoma. In these cells, studies of sphere formation indicated that OOS down-regulated stem cell proteins such as nestin, CD133, CD44, or SOX2. Nestin and CD44 are considered as markers of glioblastoma cancer stem cells, while SOX2 is a particularly important marker of stemness, and its down-regulation by OOS, together with the other markers, suggests that OOS may decrease the population of tumoral stem cells [73].

## 6. Concluding Remarks

OOS is a nutritional supplement that includes several compounds with demonstrated antitumoral properties. It is well tolerated and safe, and several clinical trials have reported its beneficial effect on the quality of life of patients. Moreover, preclinical studies have indicated that the product may increase the antitumoral characteristics of drugs and treatments commonly used in the oncology clinic.

## 7. Methodology of the Review

All papers dealing with OOS in the field of oncology, either preclinical or clinical, were analyzed. Their major conclusions are commented and the papers referenced. Information about the antitumoral activities of the components of OOS was obtained from PubMed, searching for the name of the product + cancer. Special attention was given to recent review articles on the individual components and their implication in cancer.

## Figures and Tables

**Figure 1 nutrients-12-02661-f001:**
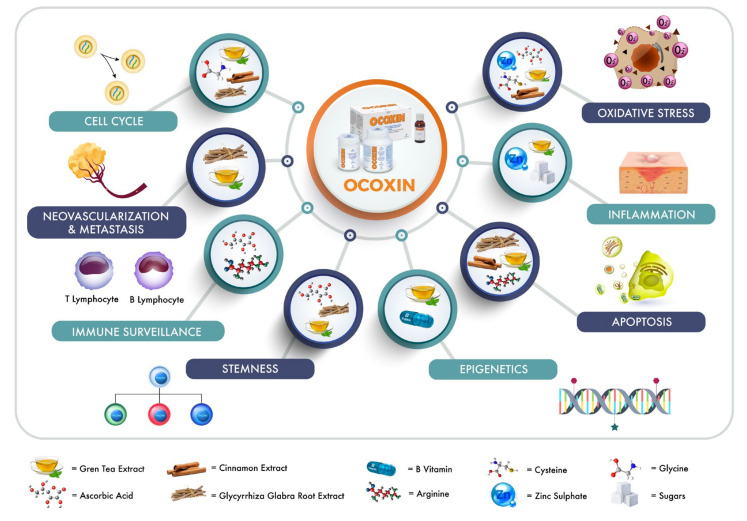
Schematic representation of biological actions of the components included in OOS formulation. The role of the different OOS components in several cellular functions such as cell cycle, vascularization and metastasis, immune surveillance, stemness, oxidative stress, inflammation, apoptosis, or epigenetics has already been described and is represented. OOS: Ocoxin Oral Solution.

**Figure 2 nutrients-12-02661-f002:**
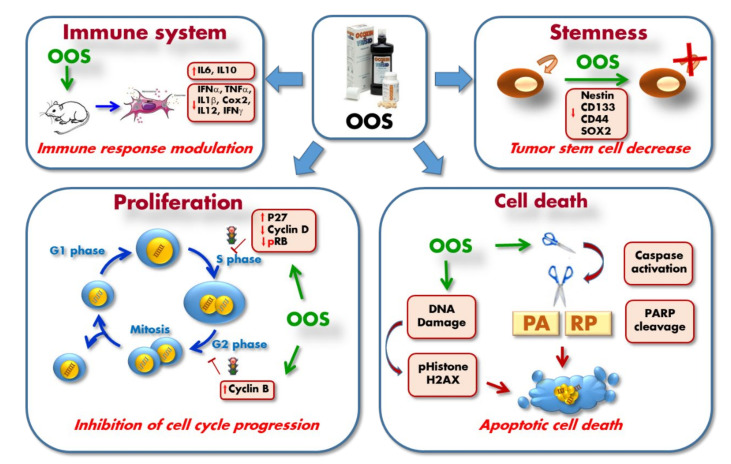
The molecular mechanism of action of OOS in several preclinical models has been described and involves regulation of cell proliferation by blockage of S phase and G2-M transition. OOS also induces caspase-dependent cell death which leads to PARP cleavage. As a consequence of the apoptotic process, DNA damage occurs and this is readout by increases in pHistone H2AX. OOS also inhibits stemness and self-renewal as indicated by the down-regulation of several stem-cell markers. Finally, an important action of OOS is related to the modulation of the immune system, as indicated by changes in the production of several cytokines. IL6: interleukin-6; IL10: interleukin 10; IFNα: interferon alpha; TNFα: tumor necrosis factor alpha; IL1β: interleukin-1 beta; Cox2: cyclo-oxygenase-2; IL12: interleukin 12; IFNγ: interferon gamma; CD133: cluster of differentiation 133; CD44: cluster of differentiation 44; SOX2: sex determining region Y-box 2; RB: retinoblastoma protein; PARP: Poly-ADP Ribose Polymerase.

**Table 1 nutrients-12-02661-t001:** List of the components included in Ocoxin (also termed Oncoxin) Oral Solution (OOS) formulation; the final amount of each component in 100 mL of the solution.

Components	Quantity
*Plant extracts*	
Glycyrrhiza glabra extract	200 mg
Green tea extract (EGC)	25 mg
Cinnamon extract	3 mg
*Vitamins*	
Ascorbic acid (Vit. C)	120 mg
Pyridoxine (Vit. B6)	4 mg
Cyanocobalamin (Vit. B12)	2 μg
Folic acid (Vit. B9)	400 μg
Calcium pantothenate (Vit. B5)	12 mg
*Amino acids*	
Glycine	2000 mg
Arginine	640 mg
Cysteine	204 mg
*Sugars*	
Glucosamine	2000 mg
Sucralose	24 mg
*Other components*	
Malic acid	1200 mg
Zinc sulfate	80 mg
Manganese sulfate	4 mg
Sodium benzoate	100 mg
Potassium sorbate	100 mg
Maracuya Aroma	50 mg

**Table 2 nutrients-12-02661-t002:** Clinical trials analyzing the effect of OOS in several cancer models and principal results obtained.

Study	Findings	Conclusions	Hospitals	Patients	Drugs for Combinations	Reference
OOS given to patients with end-stage hepatocellular carcinoma (HCC) as monotherapy.	Increased survival of patients with end-stage hepatocellular carcinoma due to intake of OOS.	Terminally ill and end stage HCC patients may be managed by food supplements such as OOS and OOS capsules.	Department of Hepatology, Bangabandhu Sheikh Mujib Medical university, Shahbagh, Dhaka, Bangladesh and Department of Medical Sciences, Toshiba General Hospital, Tokio, Japan.	29	Monotherapy with OOS and OOS capsules.	NCT 01392131 [79]
Patients with terminal stage of hepatocellular carcinoma were studied from the micronutritional point of view after OOS administration.	In patients with terminal stage HCC, OOS improved appetite, quality of life and well-being. OOS also improved overall survival in this group of patients.	The use of micronutrients and aminoacids in cancer patients undergoing chemotherapy is essential to maintain patients quality of life.	Multidisciplinary Oncology Institute, Murcia, Spain.	Not indicated.	Monotherapy with OOS.	[78]
Proof of concept study of OOS plus chemo and radiotherapy in patients with Stage IIB-III of cutaneous melanoma.	OOS showed a good safety profile in patients with stage IIB-III cutaneous melanoma. Patients kept a stable quality of life at the end of study and a high progression-free survival rate.	OOS increased progression-free survival rate in cutaneous melanoma.	Department of Dermatology and Infectious Diseases, Manuel Fajardo University Hospital, La Habana, Cuba. Department of Oncology, National Institute of Oncology (INOR), La Habana, Cuba	20	Surgery, Interferon, Paclitaxel and Temozolomide.	NCT 03541148 [80]
OOS administered with chemo and radiotherapy in gastric cancer IIB-IIIC and non-small cell lung cancer IIB-IIIA was studied from the point of view of quality of life and toxicities induced by cancer therapies.	Toxicity and side effects due to chemotherapy were diminished.	OOS helps to maintain appetite, body mass and quality of life in patients with advanced cancer treated with chemotherapy.	Kazakh Research Institute of Oncology and Radiology, Almaty, Republic of Kazakhstan. Department of Pharmacology and Clinical Pharmacology, Khanty-Mansiysk State Medical Academy, Khanty-Mansiysk, Russia. Department of Chemotherapy, Moscow Clinical Scientific Center n. a. A.S. Loginov, Moscow.	133	Xelox or paclitaxel plus carboplatin.	NCT 03550482 [75]
Phase II randomised double-blind study of patients with head and neck cancer which undergo cancer protocol therapies associated with OOS.	OOS during radiotherapy or concomitant with chemotherapy in patients with head and neck cancer, improved the quality of life and decreased the number and level of toxicities from these treatments without interfering with their mechanism of action.	OOS had a positive effect on quality of life in patients treated with radio/chemotherapy.	Department of Otolaryngology, Radiotherapy and Chemotherapy, National Institute of Oncology and Radiobiology (INOR), La Habana, Cuba.	60	Radiotherapy and Cisplatin.	NCT 03541772 [74]
OOS in the Management of chemotherapy – and or radiotherapy-associated oral mucositis.	OOS rapidly improved oral mucositis, as measured by the WHO Oral Toxicity Scale, maintained body mass and decreased the toxicity of anticancer therapy.	OOS helped to maintain normal eating habits and decreased side effects of radio/chemotherapy.	Medical Scientific Centre of Professor Shumsky, Samara, Russia. Khanty-Mansiysk Regional Hospital, Khanty-Mansiysk, Russia.	15	Monotherapy with OOS.	NCT 03577535 [76]
Findings of the 3 months supportive treatment with OOS beside the standard modalities of patients with different neoplastic diseases.	OOS along with protocol anticancer therapy improved the quality of life of patients with head and neck, breast and cervix cancer. OOS decreased episodes of depression and increased optimism. OOS reduced toxicities due to chemo and radiotherapies and increased survival rates.	The administration of OOS along with conventional chemo and radiotherapy lead to relevant anti-tumor synergy as well as to the inhibition of conventional therapy’s toxicity.	Department of Radiotherapy, Rajshahi Medical College, Rajshahi, Bangladesh.	90	Conventional radiotherapy and chemotherapy for head and neck, breast and cervix cancer.	[81]
Efficacy of OOS-VIUSID on the reduction of adverse reactions to chemotherapy and radiotherapy in patients diagnosed with cervical cancer and endometrial adenocarcinoma.	OOS-VIUSID significantly reduced the number of patients who suffered adverse events to onco-specific treatment. OOS-VIUSID stopped the fall in haemoglobin levels, and platelet and leukocyte counts, compared to patients receiving traditional treatment.	The administration of OOS along with chemo and radiotherapy lead to inhibition of conventional therapy’s toxicity and consequently less interruption of the cancer treatments.	Hospital Ramón González Coro, La Habana, Cuba. National Institute of Oncology and Radiobiology (INOR), La Habana, Cuba.	63	Radiotherapy and Cisplatin	NCT 03540407 [77]

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
