# Peer review of "Antitumoral Properties of the Nutritional Supplement Ocoxin Oral Solution: A Comprehensive Review"

_nutrients, 2020, doi:10.3390/nu12092661_

Round 1
Reviewer 1 Report
This review article mainly described and discussed the anti-tumor effects of main components, the possible mechanism of actions of Ocoxin Oral Solution (OOS). Authors also discussed the results of recent clinical trials of ONCOXIN® in several types tumors and cancers. I think this manuscript is of interest and well written. Overall, this review will be very informative for readers, particularly who are interested in novel therapeutic strategy using natural nutritional ingredients for cancer treatments. There are some concerns that should be addressed.
Major comments:
1. Authors should add more detailed description in the Table 2, including hospitals, trial periods, patient numbers, drugs for combination treatments, clinical findings, etc. Authors should discuss the current states of the clinical trials of ONCOXIN® based on the Table 2. It will be helpful for better understanding the results of the clinical trials of OOS in various cancers.
2. I strongly suggest authors to discuss what are the synergistic effects of combination of three major ingredients of OOS, Glycyrrhiza glabra extract, Green tea extract, and Cinnamon extract on tumor (cancer) inhibition, with appropriate references.
Minor comments:
1. In the Abstract, an abbreviation of ‘Ocoxin oral solution’ is required: Ocoxin oral solution (OOS).
2. In the Table 1, an abbreviation of ‘EGCG’ is required. Authors used a terminology ‘EGC’ instead of ‘EGCG’ in the text. Therefore, EGCG in the Table 1 should be changed to EGC.
3. In the Fig 1. Ascorbic acid included in an OOS is not from fruit extracts, so a cartoon of ‘fruits’ should be changed to a cartoon of ‘chemical structure of Ascorbic acid’.
4. There are two captions in the Table 2. Reference numbers should be added in the Table 2.
5. In the Figure 2, letter fonts should be changed as same as those in Figure 1. Abbreviations are needed in the Figure legend.
Author Response
This review article mainly described and discussed the anti-tumor effects of main components, the possible mechanism of actions of Ocoxin Oral Solution (OOS). Authors also discussed the results of recent clinical trials of ONCOXIN® in several types tumors and cancers. I think this manuscript is of interest and well written. Overall, this review will be very informative for readers, particularly who are interested in novel therapeutic strategy using natural nutritional ingredients for cancer treatments. There are some concerns that should be addressed.
AUTHORS’ RESPONSE: The authors wish to thank the Reviewer for these positive comments about the convenience of a review as this one on a product that uses natural nutritional components with anticancer properties.
Major comments:
- Authors should add more detailed description in the Table 2, including hospitals, trial periods, patient numbers, drugs for combination treatments, clinical findings, etc. Authors should discuss the current states of the clinical trials of ONCOXIN® based on the Table 2. It will be helpful for better understanding the results of the clinical trials of OOS in various cancers.
AUTHORS’ RESPONSE: Table 2 has been modified to include the data requested by the Reviewer. The only thing we have been unable to include are trial periods, as such information is not available in most of the papers mentioned in such table.
- I strongly suggest authors to discuss what are the synergistic effects of combination of three major ingredients of OOS, Glycyrrhiza glabra extract, Green tea extract, and Cinnamon extract on tumor (cancer) inhibition, with appropriate references.
AUTHORS’ RESPONSE: We have not found papers describing the combined antitumoral action of the three major components of OOS. However, since we consider that the study of the antitumoral effects of the combination of the three products is a good idea, we have commented it in the review (lanes 174-176).
Minor comments:
- In the Abstract, an abbreviation of ‘Ocoxin oral solution’ is required: Ocoxin oral solution (OOS).
AUTHORS’ RESPONSE: The requested change has been included in lane 9.
- In the Table 1, an abbreviation of ‘EGCG’ is required. Authors used a terminology ‘EGC’ instead of ‘EGCG’ in the text. Therefore, EGCG in the Table 1 should be changed to EGC.
AUTHORS’ RESPONSE: The requested change has been included in lane 43.
- In the Fig 1. Ascorbic acid included in an OOS is not from fruit extracts, so a cartoon of ‘fruits’ should be changed to a cartoon of ‘chemical structure of Ascorbic acid’.
AUTHORS’ RESPONSE: The requested change has been included in Figure 1. Now the figure shows the chemical structure of Ascorbic acid instead of the picture of fruits.
- There are two captions in the Table 2. Reference numbers should be added in the Table 2.
AUTHORS’ RESPONSE: A new version of table 2 with the additional data requested by this Reviewer, including reference numbers, has now been prepared.
- In the Figure 2, letter fonts should be changed as same as those in Figure 1. Abbreviations are needed in the Figure legend.
AUTHORS’ RESPONSE: We have corrected this in figure 2. The font type used in both figures is now the same (Futura font). Abbreviations have been included in the legend of figure 2 (lanes 433-436).
Reviewer 2 Report
The main problem with this paper is that there is not much to review. The authors belong to the group which is the only group publishing about this formulation. Therefore the authors are reviewing not the formulation but only the components of the formulation. Biological activity of the several plant extracts used in the formulation are reviewed regularly (e.g. DOI: 10.3390/biom10030352) and it is not necessary to provide basic review once again. The rest of the compounds are ordinary and doesn't need review at all (glucosamine?). The results of the papers actually concerned with the formulation are interesting but also produced mainly by the people which are commercially interested.
Author Response
The main problem with this paper is that there is not much to review. The authors belong to the group which is the only group publishing about this formulation. Therefore the authors are reviewing not the formulation but only the components of the formulation. Biological activity of the several plant extracts used in the formulation are reviewed regularly (e.g. DOI: 10.3390/biom10030352) and it is not necessary to provide basic review once again. The rest of the compounds are ordinary and doesn't need review at all (glucosamine?). The results of the papers actually concerned with the formulation are interesting but also produced mainly by the people which are commercially interested.
AUTHORS’ RESPONSES: The Reviewer questions the real value of this review. We wish to indicate that this is the first review prepared on Ocoxin. This product is increasingly being used, and since no comprehensive reviews exist, we think that the one we prepared is timely and adequate. Moreover, the value of the review is underlined by Reviewers 1 and 3. We consider that this review, being the first comprehensive one on OOS, will be of interest and value for the professionals that recommend the use of OOS. We also wish to indicate that not all authors have participated in all the papers dealing with OOS. In fact, AP and EDR have not participated in any of the clinical papers, and are authors in some (but not all) preclinical ones. On the other side, we do recognize that our participation in the preclinical (AP, EDR and ES) and clinical (ES) papers is a positive fact as we very much know the OOS field in the molecular and clinical oncology area. With respect to the comment of the Reviewer on the fact that the authors are commercially interested in the product, please note that we have clearly included information about funding and conflicts of interest in the paper, including affiliation of ES to the company that distributes the product.
Reviewer 3 Report
Atanasio et al described the detailed role of Oncoxin (Oral drug) and its nutrietional supplement in relevant to its anti-tumoral effects in this exquisite comprehensive review. This drug has been extensively studied by the same group and going with phase II clinical trial sponsored by Catalysis SL (Clinicaltrails.gov/NIH). It is very much essential to spread the consciousness of this drug with the extensive details and mechanism of action. Atanasio et al., encompasses all essential aspects in this comprehensive review to reach the scientific community as well as the laypersons. Authors need to look for minor spell check and grammatical errors.
Author Response
Atanasio et al described the detailed role of Oncoxin (Oral drug) and its nutrietional supplement in relevant to its anti-tumoral effects in this exquisite comprehensive review. This drug has been extensively studied by the same group and going with phase II clinical trial sponsored by Catalysis SL (Clinicaltrails.gov/NIH). It is very much essential to spread the consciousness of this drug with the extensive details and mechanism of action. Atanasio et al., encompasses all essential aspects in this comprehensive review to reach the scientific community as well as the laypersons. Authors need to look for minor spell check and grammatical errors.
AUTHORS’ RESPONSES: We wish to thank the Reviewer for the positive comments regarding the manuscript. Those comments recognize its value and the need for such a timely review on Ocoxin, due to its increasing use particularly in the oncology clinic.
With respect to the spelling and grammar corrections, we have carefully revised the paper and made some corrections. Perhaps we may have missed some, so if the Reviewer can be more precise we will be very happy to correct them and very thankful.